# Molecular identification of *Coxiella burnetii* in raw milk samples collected from farm animals in districts Kasur and Lahore of Punjab, Pakistan

Shahpal Shujat[1,2], Wasim Shehzad[1], Aftab Ahmad Anjum[3], Julia A. Hertl[2], Yrjö T. Gröhn[2], Muhammad Yasir Zahoor[1]*

1 Institute of Biochemistry and Biotechnology, University of Veterinary and Animal Sciences, Lahore, Punjab, Pakistan, 2 Department of Population Medicine and Diagnostic Sciences, College of Veterinary Medicine, Cornell University, Ithaca, New York, United States of America, 3 Institute of Microbiology, University of Veterinary and Animal Sciences, Lahore, Punjab, Pakistan

* yasir.zahoor@uvas.edu.pk

## Abstract

*Coxiella burnetii* is the worldwide zoonotic infectious agent for Q fever in humans and animals. Farm animals are the main reservoirs of *C. burnetii* infection, which is mainly transmitted via tick bites. In humans, oral, percutaneous, and respiratory routes are the primary sources of infection transmission. The clinical signs vary from flu-like symptoms to endocarditis for humans' acute and chronic Q fever. While it is usually asymptomatic in livestock, abortion, stillbirth, infertility, mastitis, and endometritis are its clinical consequences. Infected farm animals shed *C. burnetii* in birth products, milk, feces, vaginal mucus, and urine. Milk is an important source of infection among foods of animal origin. This study aimed to determine the prevalence and molecular characterization of *C. burnetii* in milk samples of dairy animals from two districts in Punjab, Pakistan, as it has not been reported there so far. Using a convenience sampling approach, the current study included 304 individual milk samples from different herds of cattle, buffalo, goats, and sheep present on 39 farms in 11 villages in the districts of Kasur and Lahore. PCR targeting the *IS1111* gene sequence was used for its detection. *Coxiella burnetii* DNA was present in 19 of the 304 (6.3%) samples. The distribution was 7.2% and 5.2% in districts Kasur and Lahore, respectively. The results showed the distribution in ruminants as 3.4% in buffalo, 5.6% in cattle, 6.7% in goats, and 10.6% in sheep. From the univariable analysis, the clinical signs of infection i.e. mastitis and abortion were analyzed for the prevalence of *Coxiella burnetii*. The obtained sequences were identical to the previously reported sequence of a local strain in district Lahore, Sahiwal and Attock. These findings demonstrated that the prevalence of *C. burnetii* in raw milk samples deserves more attention from the health care system and veterinary organizations in Kasur and Lahore of Punjab, Pakistan. Future studies should include different districts and human populations, especially professionals working with animals, to estimate the prevalence of *C. burnetii*.

**Data Availability Statement:** All relevant data for this study are publicly available from the OSF repository (https://osf.io/hzd9c).

**Funding:** Funding Statement This study was funded by International Research Support Initiative Program, Higher Education Comission, Pakistan fellowship. Grant: I -8/HEC/HRD/2021/11529 awarded to student. The funder has no role in study design, data collection and analysis, decision to publish, or preparation of manuscript.

**Competing interests:** Competing interests: The authors have declared that no competing interests exists.

## Introduction

*Coxiella burnetii* is a gram-negative obligate intracellular bacterium. It is an etiological agent of Q fever in humans and animals [1]. The Center for Disease Control and Prevention has classified it as a category B bioweapon due to its aerosolized property and rapid spread. This bacterium can cause disease in a low infectious dose; only 1–10 organisms are required for this infection [2–4]. Its spore-like structure can persist under harsh environmental conditions and stress [5].

*Coxiella burnetii* has been globally ranked among the top 13 priority zoonotic pathogens. It can infect different host species, including domestic, wild, and marine mammals, birds, reptiles, and arthropods. Ruminants are the main reservoirs for this pathogen [6]. It is a tick-borne pathogen; thus, it is transmitted to ruminants mainly through ticks [7]. In humans, infection may occur through inhalation of particles dispersed from environmental dust [8] and direct contact with contaminated milk, meat, urine, semen, and feces [9]. Clinical manifestation of this bacterium in ruminants includes stillbirth, abortion, mastitis, endometritis, and other reproductive disorders [10,11]. In humans, flu, fever, hepatitis, and endocarditis are the main manifestations [12].

The pathogen was first identified in Australian abattoir workers [13]. It has been considered endemic and has a worldwide distribution including Pakistan. It has gained international attention since the outbreak in the Netherlands from 2007 to 2010, and has affected humans and farm animals in other European countries [11]. Most cases remain undiagnosed due to a lack of proper diagnostic tools in developing countries like Pakistan.

It has been detected using serological and molecular tests from numerous samples, i.e., blood, serum, milk, and meat. Culturing techniques for *C. burnetii* detection are rarely used because of its high pathogenicity. PCR is the most sensitive molecular technique for the detection of *C. burnetii*. Single copy and multicopy gene targets are used for its detection. The superoxide dismutase gene and *IS1111* gene are single and multicopy gene targets [14].

Raw milk from livestock animals is a potential source of *C. burnetii* infection. In the USA, there was a report of infection by *C. burnetii* by consumption of raw milk from a dairy in Michigan [15] and a further study reported *C. burnetii* infection in Colombian farmers [16]. Although seroconversion is frequently the result of consuming raw milk or dairy products, clinical Q fever is rarely a consequence [17]. Food has been considered a "seldom recorded route" for the transmission of *C. burnetii*; at this point, food consumption cannot be eliminated from the assessment of Q fever transmission channels, and a One Health approach must extend to consideration of this potential. Simulation studies conducted by Gale et al. [18], together with a comprehensive literature review in 2018 by Pexara et al. [19] led the latter authors to conclude that the risk of *C. burnetii* human infection due to consumption of unpasteurized milk and raw milk products "cannot be considered negligible" [18,19].

The bacterium has been known to shed via the milk of cattle in several European countries; in Poland, for example, it was found in 31.54% of tested dairy cattle herds' milk [20]. In Latvia, a survey of dairy cattle operations revealed that 10.7% of bulk tank milk samples tested positive for the presence of *C. burnetii* DNA [21].

Several countries have tested dairy products other than milk for the presence of *C. burnetii* DNA. It was discovered that 69.2% of the dairy products in Poland tested positive [20]. In Spain, 7.6% of hard cheeses made from raw sheep's milk included pathogenic bacteria, and 29.9% of them had *C. burnetii* DNA [22].

*Coxiella burnetii* is a neglected pathogen in Pakistan, although it strongly impacts an infected country's economy and public health. Appropriate farm management and public

awareness are required to control this infection. Moreover, the infection remains largely undetected mainly due to limited diagnostic facilities and the lack of sufficient training to healthcare workers and clinical physicians for this contagious disease in developing countries like Pakistan. Notably, in the previous 65 years, there have been only six publications on human and animal Q fever from Pakistan in the PubMed databank [23–29]. Information regarding *C. burnetii*'s manifestation in milk obtained from small and large ruminants for human consumption has not been assessed so far in Pakistan, although there are reports from other countries in Asia including India and China. The objective of the current study was to estimate the prevalence and molecular characterization of *C. burnetii* in milk samples collected from ruminants used for human consumption.

## Methodology

### Study area and sampling

The sampling was conducted in 2019 with the help of the livestock department of Districts Kasur and Lahore. The samples were collected from the consent of dairy farmers using a convenience approach (Fig 1). The study included 304 individual milk samples of 90 cattle, 88 buffalo, 60 goat, and 66 sheep collected from 39 farms in 11 Kasur and Lahore villages. Milk samples of about 5 ml were collected in 15 ml sterile falcon tubes, and the information was recorded in the sample collection data book. The collected samples were transported at 4 to 8˚C temperature.

### Inclusion and exclusion criteria

The local commercial and household farms were selected in the current study while institutional farms were excluded from the study. Animals that were treated with antibiotics or diagnosed with other diseases were excluded from the study.

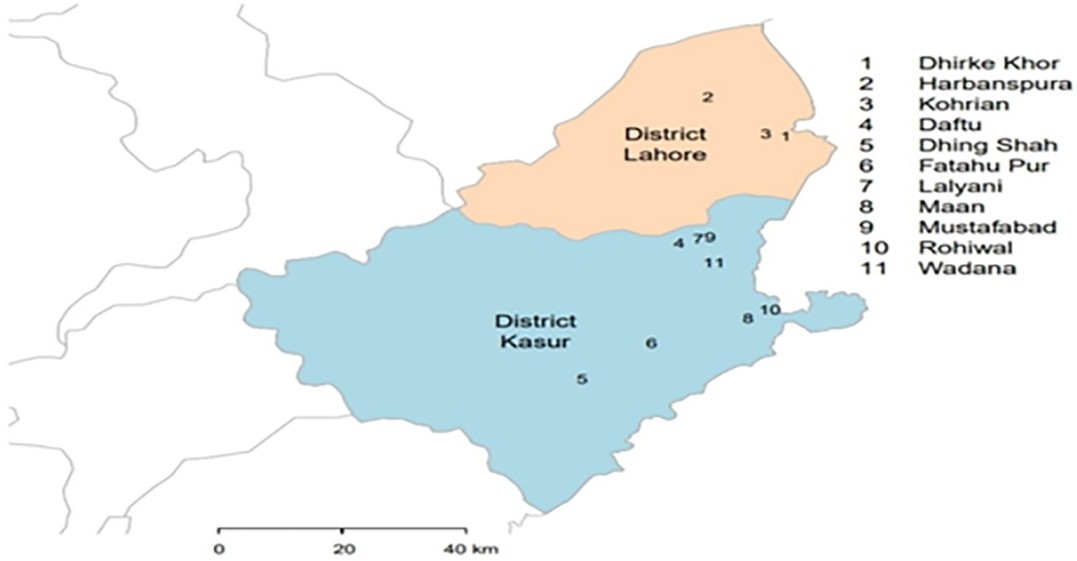

**Fig 1. A map showing the sampling locations in the study area of districts Kasur and Lahore, Punjab.** The map was created using GADM R-package, developed by [30].

## Milk processing and DNA isolation

Milk samples were stored at -20˚C and further processed for DNA extraction using a manual method. Milk samples of 200 µl were placed in a microcentrifuge tube and centrifuged at 14500 rpm for 15 min at 4˚C, and the cream layer was separated [31]. Lysis buffer of 700 µl and 10 µl of proteinase K were taken in its pellet and incubated at 56˚C overnight. After overnight incubation, 500 µl of PCI was added and vortexed until the solution turned milky. Then it was centrifuged under identical conditions as mentioned above. Three layers were formed, and the upper transparent layer containing DNA was taken into a separate microcentrifuge tube. Two parts of isopropanol and 200 µl of chilled absolute ethanol were added in 1 part aqueous transparent layer and incubated for 20 min at -20˚C. It was then centrifuged under the same conditions, the supernatant was discarded, and the pellet was taken. The taken pellet was washed using 200 µl of 70% ethanol and centrifuged under the same conditions. Then the supernatant was discarded, leaving the pellet drying overnight to evaporate ethanol, and acting as a PCR inhibitor. The dried pellet was dissolved in 20 µl of distilled water in a water bath and heat shocked at 70˚C for 40 minutes. DNA quality and quantity were assessed using a spectrophotometer.

## Molecular assay and sequence analysis

*Coxiella burnetii* was diagnosed using multiple copy gene amplification assays targeting transposase gene, i.e., *IS1111*; specific primers were used for this assay. The set of primers used for the PCR amplification assay was sequenced as 5'-GTCTTAAGGTGGGCTGCGTG-3'and 5'-CCCCGAATCTCATTGATCAGC-3' for forward and reverse primer [32]. The assay was sensitive due to its multiple copy gene targets and specificity was confirmed by alignment with the reported sequences of *C. burnetii* and results were validated using alignment of the obtained sequences for positive samples. The diagnostic assay was validated using Vircell Amplirun® Coxiella DNA Control. Each PCR reaction test contained 12.5 µl of 2X master mix, 1.25 µl of 10µM forward and reverse primer, and 1 µl of 50–100 ng DNA in a final volume up to 25 µl by adding nuclease-free water. PCR reaction was performed using a 96 well Applied Biosystems by Thermo Fisher Scientific thermal cycler. The conditions for reaction were optimized as initial denaturation at 95˚C for 5 min, final denaturation at 94˚C for 30 sec, annealing at 60˚C for 30 sec, extension at 72˚C for 1 min, repeat steps 1 to 3 for 30 cycles, and final extension at 72˚C for 10 min. PCR products were analyzed on 2% agarose gel, and specific product was identified, i.e., 294 bp was observed during analysis. The positive samples were sequenced from commercially available services. The sequences were analyzed for phylogenetics using the MEGA version 6.0 bioinformatics tool. Alignment and phylogenetic tree construction of 12 sequences, including two query sequences, were performed using the MEGA tool by the maximum likelihood method [33].

## Data analysis

The data were recorded in a Microsoft Excel spreadsheet. The analysis was performed using SAS 9.4 statistical package (see S1 File). Fisher exact tests were performed using the fisher.test function in R 4.3.2, and logistic regression models were fitted using PROC LOGISTIC in SAS 9.4 with the presence/absence of *C. burnetii* DNA as the outcome variable. The explanatory variables in the study included species, district and abortion.

## Ethics statement

The University of Veterinary and Animal Sciences' Advanced Studies and Research Board granted ethics approval to undertake the study in its 50th meeting held on 8 February 2019.

**Table 1. Prevalence of *Coxiella burnetii* in milk samples collected from different species by PCR during 2019, from districts Lahore and Kasur, Pakistan.**

|            | No. of samples examined | No. of positive samples | Percentage |
|------------|------------------------|------------------------|------------|
| **Species** |                        |                        |            |
| Cattle     | 90                     | 5                      | 5.6        |
| Buffalo    | 88                     | 3                      | 3.4        |
| Goat       | 60                     | 4                      | 6.7        |
| Sheep      | 66                     | 7                      | 10.6       |
| **District** |                      |                        |            |
| Lahore     | 136                    | 7                      | 5.2        |
| Kasur      | 168                    | 12                     | 7.2        |
| Total      | 304                    | 19                     | 6.3        |

The raw milk samples and information collected from the individual animals was collected with the informed consent of dairy farmers. The raw milk samples were collected with the assistance of animal handlers.

## Results

Of 304 samples, 19 (6.3%) were positive for *C. burnetii* DNA by PCR detection using the *IS1111* sequence. The observed results showed a higher prevalence of *C. burnetii* in district Kasur (7.1%) when compared with district Lahore (5.1%) (Table 1), although it was not statistically significant (P = 0.63). *Coxiella burnetii* prevalence in the four species of ruminants was 3.4% in buffalo, 5.6% in cattle, 6.7% in goats, and 10.6% in sheep (Table 1). Thus, the prevalence of *C. burnetii* was higher in milk samples obtained from goats and sheep than in cattle and buffalo, although the differences between districts for each species were not statistically different (Table 2). From a logistic regression model with district, species, and an interaction between district and species, in Lahore, sheep were 13 times (95% confidence limits: 1.36, 126.02) more likely to test positive than were buffalo, and were 10.4 times (95% confidence limits: 1.07, 100.0) more likely to test positive than were cattle. No significant association was detected between abortion and infection status for any of the species (Table 3).

When the sequences were aligned on the basis of their origin, the results showed that in the current study, the sequences obtained from district Lahore were clustered with previously reported sequences from districts Lahore and Sahiwal because of their close geographical proximity. At the same time, the sequences obtained from district Kasur were clustered separately with previously reported sequences from district Attock (Fig 2).

## Discussion

The aims of the present study were to estimate the prevalence and undertake a molecular characterization of *C. burnetii* DNA in raw milk samples collected from different herds of

**Table 2. Prevalence of *Coxiella burnetii* in milk samples collected during 2019 from different species and its distribution in districts Lahore and Kasur, Pakistan.**

| Species | District Lahore | District Kasur | Overall | P value |
|---------|----------------|----------------|---------|---------|
| Cattle  | 2.5% (1/40)    | 8% (4/50)      | 5.6% (5/90) | 0.30 |
| Buffalo | 2% (1/50)      | 5.3% (2/38)    | 3.4% (3/88) | 0.37 |
| Goat    | 3.7% (1/27)    | 9.1% (3/33)    | 6.7% (4/60) | 0.39 |
| Sheep   | 21.1% (4/19)   | 6.4% (3/47)    | 10.6% (7/66) | 0.34 |

**Table 3. Prevalence of *Coxiella burnetii* in milk samples collected from different species when the clinical sign, abortion was analyzed using univariate analysis.**

| Abortion | Infection | Cattle | Buffalo | Goat | Sheep | Overall |
|---|---|---|---|---|---|---|
| | | Frequency | Frequency | Frequency | Frequency | Frequency |
| Yes | Positive | 1 (33%) | 0 | 0 | 0 | 1 |
| | Negative | 2 (67%) | 0 | 2 | 0 | 4 |
| | Total | 3 | 0 | 2 | 0 | 5 |
| No | Positive | 4 (5%) | 3 (3%) | 4 | 7 | 18 (6%) |
| | Negative | 83 (95%) | 85 (97%) | 54 | 59 | 281 (94%) |
| | Total | 87 | 88 | 58 | 66 | 299 |
| P value | | 0.16 | - | - | - | 0.28 |

ruminants in the districts of Lahore and Kasur, Pakistan. The first evidence of *C. burnetii* DNA in milk was obtained during this study in Pakistan. Previous studies evaluated the pathogen's presence in meat, ticks, blood, and serum samples collected from ruminants, environmental samples (i.e., soil samples obtained from animal farms), and human blood samples [22–28]. *Coxiella burnetii* in raw milk from ruminants was not studied previously. The absence of previous records was due to a lack of awareness and neglect of the disease caused by this pathogen.

There is variation in the clinical manifestations of *C. burnetii* in humans. In general, it may be asymptomatic or may show signs of acute or chronic manifestation [34]. People with acute *C. burnetii* infection typically experience mild or asymptomatic flu-like symptoms [1]. Most of the time, only a small percentage of infected individuals exhibit an actual disease, which can lead to serious complications [11]. The disease may cause a chronic infection in certain infected cases, which could have long-term sequelae. *C. burnetii* was thought to be primarily an occupational hazard, and human infection may be strongly related to direct contact with ruminants [16]. Furthermore, an assessment of the zoonotic pathogen presence within foods of animal origin and the potential threat to public health is required considering the high prevalence of *C. burnetii* in production animals. When it pertains to foods of animal origin, raw milk is the most important source of *Coxiella burnetii* infection, through raw milk or raw milk products [17]. The current study was based on a one health approach and this zoonotic pathogen was recovered from milk samples used for human consumption. The estimated overall prevalence of *C. burnetii* was 6.25%, while it was 7.14% in district Kasur and 5.15% in district Lahore. Previous studies suggested a 36.87% *C. burnetii* prevalence in blood samples of small ruminants from district Kasur while the prevalence of *C. burnetii* was estimated at 32.1% and 12.5% in cattle and buffaloes, respectively [35]. For district Lahore, the prevalence of infection was 4.8% in environmental samples from a previously reported study [28].

Based on the results of this study, *C. burnetii* was detected in 6.7% of goat milk samples by PCR using the *IS1111* gene sequence. The prevalence of *C. burnetii* in districts Lahore and Kasur was 2% and 9%, respectively. A previous study estimated the prevalence of infection in goat blood samples as 30% in district Kasur [35]. The reported prevalence of infection in other countries was 6.3–12.1% in Belgium [36], 14.3% in the USA [37], and 17.2% in Lebanon [38]. Goat's milk is frequently consumed in various countries throughout the world. Therefore, goat milk used for human consumption should be screened for *C. burnetii*, among other pathogens.

The results showed that 10.6% of sheep milk samples were positive for the causative agent of Q fever. The estimated prevalence in the two districts was 21.1% and 6.4% in Lahore and Kasur, respectively. In another recent study, *C. burnetii* prevalence was reported as 46.9% in

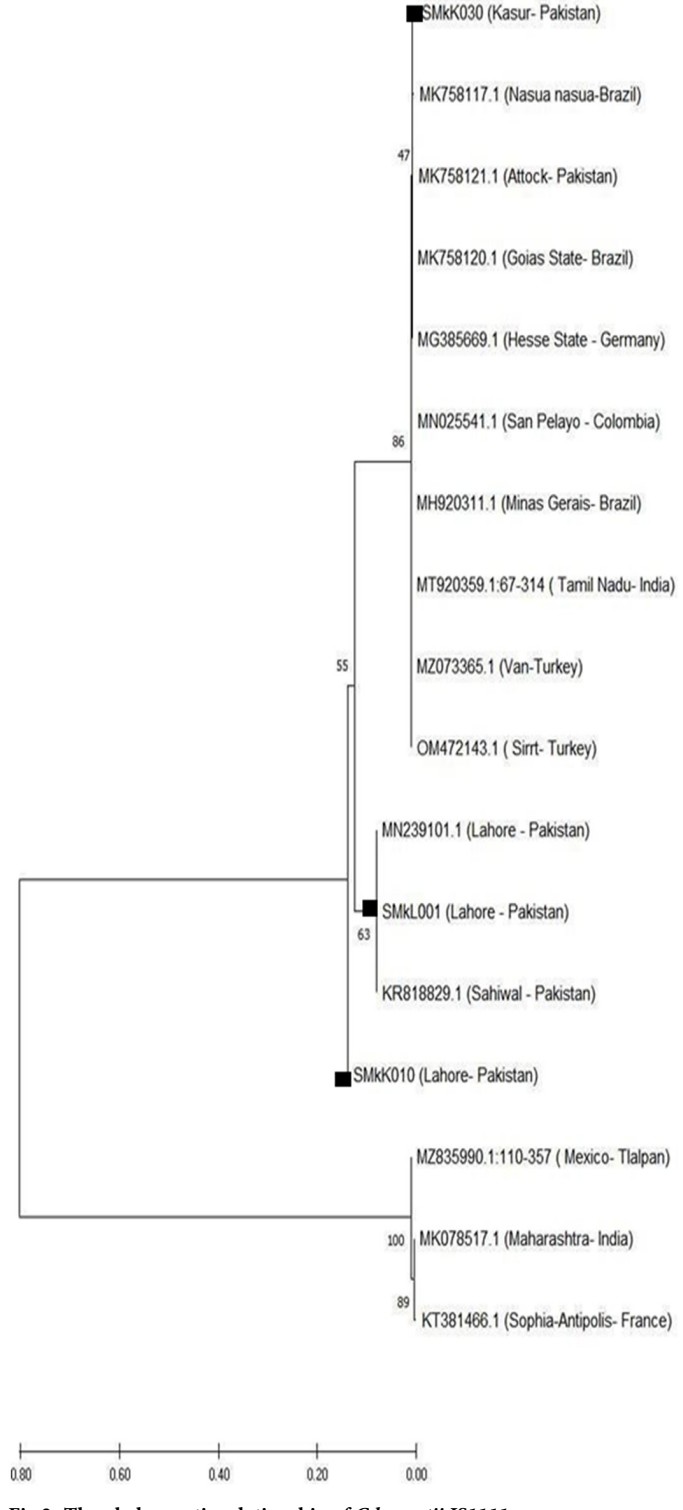

**Fig 2. The phylogenetic relationship of *C.burnetii IS1111* gene sequence recovered from milk samples in 2019 in districts Lahore and Kasur.** Query sequences are labeled in the figure.

the Kasur district [39]. The prevalence of *C. burnetii* in other countries was 10% in Lebanon [38], 4% in Hungary [40], 6.5% in Turkey [41], and 22% in Spain [42]. According to the findings of this study and previous studies in Pakistan and other countries, it seems that *C. burnetii* is common in sheep. Like goat milk, there is a strong interest in consuming raw sheep's milk and its products in Pakistan, especially in rural and nomadic populations that consume milk from these species. Therefore, paying attention to milk-borne pathogens in such communities is essential, and veterinary organizations must prioritize control and prevention strategies in livestock.

The current study estimated *C. burnetii* prevalence in cattle as 5.6%; the prevalence was 2.5% in Lahore district and 8% in Kasur district. The prevalence of *C. burnetii* in buffaloes was 3.4%. The distribution of *C. burnetii* prevalence in buffaloes in Lahore and Kasur was estimated at 2% and 5.3%, respectively. A previous study showed 32.12% and 12.5% in cattle and buffaloes for the prevalence of *C. burnetii* in district Kasur [39]. Different prevalences of *C. burnetii* have been reported in cattle milk from other countries: 8.7% in Hungary [40], 15.1% in Lebanon [38], 18.8% in the Netherlands [43] and 27% in Italy [44]. Therefore, shedding of pathogens in milk by bovines appears to be the most critical route of spreading this bacterium in the environment in all investigated countries. Future studies should include other districts of Punjab, Pakistan, and sectors of the human population at risk, especially professionals, i.e., farm workers and veterinarians. Given that some countries have reported pediatric cases of alimentary infection with *C. burnetii* and that dairy products are a staple of the diet starting at six months of age, special attention was given to yoghurts and other fermented products with added ingredients that are frequently consumed by both adults and children [45,46]. In this study, molecular evidence of *C. burnetii* was detected in milk samples of dairy animals in the districts of Kasur and Lahore. These findings demonstrated that *C. burnetii* prevalence, especially in raw milk samples, could pose a severe risk of Q fever to farm workers and consumers in Punjab, Pakistan.

One limitation of the current study is the relatively small number of cases detected, and this limited the statistical power to detect associations such as differences in prevalence between species and between districts. This could be addressed by further monitoring of milk samples, not only from these two districts, but in other districts and across the provinces of Pakistan. In addition, further molecular characterization and identification of other gene targets to identify strains would be helpful in the context of Pakistani dairy animals. Nevertheless, it is considered an important finding about *Coxiella* infection of milk products in Pakistan, and it is recommended that the dairy industry should implement screening of dairy items manufactured from raw milk such as cream, desserts, yoghurt, butter and cheese to address this public health issue.

## Conclusion

Molecular evidence of *C. burnetii* was observed in milk samples of cattle, buffalo, goats, and sheep collected from the farm in two districts of Punjab, Pakistan. These findings emphasized that the prevalence of *C. burnetii*, especially in raw milk samples, deserves more attention from the public health system and dairy industry in Pakistan. The stakeholders in the dairy sector should be attentive for diagnosis and vaccination of dairy animals for *C. burnetii* infection. The infection causes abortion, mastitis and other reproductive issues in dairy animals, and the milk from the animals with mastitis cannot be sold. The abortion and other reproductive issues in dairy animal due to *C.burnetii* infection are important in economic and health aspects. Furthermore, dairy products should market the pasteurized milk and milk products to ensure adequate consumer health protection. Future studies must include other districts and

risk evaluation in the human population for the infection, especially in professionals, i.e., farm workers, dairy processing industry workers and veterinarians.

## Supporting information

**S1 File. Supplementary tables.**
(PDF)

**S2 File.**
(MAS)

**S3 File.**
(SAS7BDAT)

**S4 File.**
(TXT)

## Acknowledgments

The authors are grateful to Assistant Director Livestock District Kasur Dr. Musarat, Veterinary Assistants in Livestock Department District Kasur, farmers and Assistant Director Livestock, District Lahore, and Veterinary Assistants Livestock Department, District Lahore for assisting in sample collection and data recording.

## Author Contributions

**Conceptualization:** Shahpal Shujat.

**Data curation:** Shahpal Shujat, Wasim Shehzad, Aftab Ahmad Anjum.

**Formal analysis:** Shahpal Shujat, Julia A. Hertl, Yrjö T. Gröhn.

**Investigation:** Shahpal Shujat.

**Methodology:** Shahpal Shujat.

**Supervision:** Yrjö T. Gröhn, Muhammad Yasir Zahoor.

**Writing – original draft:** Shahpal Shujat.

**Writing – review & editing:** Yrjö T. Gröhn.

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
