## [Decision Letter · Decision Letter 0]

4 Jan 2024

PONE-D-23-27370Molecular identification of Coxiella burnetii in raw milk samples collected from farm animals in districts Kasur and Lahore of Punjab, PakistanPLOS ONE

Dear Dr. Yasir Zahoor,

Thank you for submitting your manuscript to PLOS ONE. After careful consideration, we feel that it has merit but does not fully meet PLOS ONE’s publication criteria as it currently stands. Therefore, we invite you to submit a revised version of the manuscript that addresses the points raised during the review process.

We look forward to receiving your revised manuscript.

Kind regards,

Gianmarco Ferrara, PhD, MVD

Academic Editor

PLOS ONE

Journal Requirements:

https://www.hindawi.com/journals/ijmicro/2021/6632036/

http://erepository.uonbi.ac.ke/bitstream/handle/11295/101328/Thesis%20final%20.pdf?isAllowed=y&sequence=1

https://journals.plos.org/plosone/article?id=10.1371%2Fjournal.pone.0289944

In your revision ensure you cite all your sources (including your own works), and quote or rephrase any duplicated text outside the methods section. Further consideration is dependent on these concerns being addressed.

4. We note that Figure 1 in your submission contain map image which may be copyrighted. All PLOS content is published under the Creative Commons Attribution License (CC BY 4.0), which means that the manuscript, images, and Supporting Information files will be freely available online, and any third party is permitted to access, download, copy, distribute, and use these materials in any way, even commercially, with proper attribution. For these reasons, we cannot publish previously copyrighted maps or satellite images created using proprietary data, such as Google software (Google Maps, Street View, and Earth). For more information, see our copyright guidelines: http://journals.plos.org/plosone/s/licenses-and-copyright.

Reviewers' comments:

Reviewer's Responses to Questions

**Comments to the Author**

1. Is the manuscript technically sound, and do the data support the conclusions?

Reviewer #1: Partly

Reviewer #2: Partly

Reviewer #3: Yes

2. Has the statistical analysis been performed appropriately and rigorously? 

Reviewer #1: No

Reviewer #2: Yes

Reviewer #3: No

3. Have the authors made all data underlying the findings in their manuscript fully available?

Reviewer #1: Yes

Reviewer #2: Yes

Reviewer #3: Yes

4. Is the manuscript presented in an intelligible fashion and written in standard English?

Reviewer #1: No

Reviewer #2: Yes

Reviewer #3: Yes

5. Review Comments to the Author

Reviewer #1: The manuscript describes the detection of Coxiella burnetii in milk samples collected from farms animals in districts Kasur and Lahore of Punjab, Pakistan. A PCR assay was performed. Unfortunately, too much information is lacking in the manuscript, particularly in Materials and Methods. My specific comments on the manuscript are shown in the attached file (highlighted in pink). Below are my other concerns:

1. For easier review, line numbers should be shown on the left side of each page. Also, please make sure to subject the manuscript to language review and editing.

2. In the introduction, please don't begin every sentence with C. burnetii, and the way punctuation marks are used confuse the reader

3. Tables in general, including the Supplementary ones are poorly organized and difficult to understand. I can't see the difference between table 1 and 2. And as a recomendation, when there are rows that add up to zero, it means that the sample size was insufficient, and that the analyzes carried out are not appropriate.

4. Phylogenetic analysis is insufficient. More isolates, both reported and those derived from their own study should be added.

5. The discussion is also poorly written. What was already mentioned in the summary, introduction and results is constantly repeated.

Reviewer #2: While the study addresses an important public health concern, there are several shortcomings and scientific mistakes that need attention.

Introduction:

The introduction is informative, but it lacks a clear statement of the research gap and a hypothesis. The authors could improve the introduction by providing more context on the global prevalence of C. burnetii in raw milk and its impact on public health. Furthermore, the reference citations in the introduction are somewhat outdated, and more recent literature should be incorporated to support the study's rationale.

Methodology:

The methodology section lacks details on the inclusion/exclusion criteria for farms and animals, potentially introducing selection bias. The use of a convenient sampling approach may not be representative of the overall population, and the authors should discuss the potential limitations associated with this sampling strategy. Additionally, the absence of information on the sensitivity and specificity of the PCR assay used raises concerns about the reliability of the results.

Discussion:

While the discussion compares the study findings with previous research, it fails to discuss the implications of the results for public health and the dairy industry in Punjab. Additionally, the discussion could benefit from a more critical analysis of the study limitations, such as the potential for contamination during sample collection and the use of a single gene target for detection.

Conclusion:

The conclusion provides a succinct summary, but it lacks specific recommendations for future research or practical implications for stakeholders. A more detailed discussion of the policy and public health implications of the study results would enhance the conclusion.

Overall, the manuscript should undergo thorough proofreading and editing for language and grammar issues. Additionally, the authors should consider updating references to include more recent publications in the field. Lastly, the manuscript would benefit from a more explicit statement on ethical considerations, especially regarding animal welfare and informed consent for sample collection.

Reviewer #3: General Comments:

The paper by Shujat et al. provided some interesting results of detection of Coxiella in milk samples of Pakistan. I understand this is the first study of its kind in Pakistan, and I think the results would be applicable to many other low-income countries with similar agriculture and dairy systems.

There were two aspects investigated: (1) molecular characterising and (2) prevalence estimation, investigating differences between species and geographic districts. The second part was a bit more restricted, in terms of sufficient number of animals across all species and district combinations, but nevertheless this is useful as a reference base for other future studies, and the authors provided a comprehensive analysis of the available data. One point that needs to be clarified is the overall objective due to a clash between the Introduction and Discussion. I think it really is both points above. In addition, some of the analyses need to be described in a bit more detail in the “Data Analysis” section to clarify some of the results being presented.

The following is a list of points I noted as I read through the manuscript. Most are minor editorial, but some may take some more time to work through. Please consider each point carefully, particularly the more substantial points, as this will result in a considerably improved manuscript.

Specific Comments:

Page 1: In the title page, I believe the affiliation for Wasim Shehzad should be ‘1’, not ‘2’.

Page 2, Abstract: (1) I think “convenience sampling” rather than “convenient”. (2) Also, be consistent in number of decimal places, suggest only one, so “7.4%” and “5.2%”. (3) Change “univariate” to “univariable” (The former indicates only one outcome variable. The latter indicates only one explanatory variable in the model). (4) Change “was differed” to “differed”.

, Paragraph 3: Suggest “contaminated milk” rather than “contagious milk”.

Page 3, Paragraph 2: Change “Multicopy” to “multicopy”.

, Paragraph 3: Change “about six” to “six” as that is what you have reported. However, please be more specific about “international databank”, e.g. a PubMed search? Make sure this list is current.

, Paragraph 4: Change “convenient” to “convenience sampling”. Please provide a breakdown of the 304 samples by species.

, Paragraph 5: Change “-20 °C temperature” to “-20 °C”.

Page 4, Paragraph 3: For the logistic regression, you need to mention the terms in the model, i.e. explanatory variable(s), not just the outcome variable. Also, where you used a chi-square analysis, is that what you labelled “univariate” (or “univariable” as mentioned above? Please clarify.

, Paragraph 4: Perhaps write as “The University of Veterinary and Animal Sciences Advanced Studies and Research Board approved the study at its 50th meeting held on 8 February 2019” (suggesting explicit month name because of differing UK / US date format conventions – or was it “2 August 2019”?

Page 5, Figure 1: Change to “Lahore” in caption.

, Paragraph 1: (1) As in Abstract, please round to 1 d.p., as sample sizes don’t lend to 2 d.p. accuracy. After “not statistically significant’, add P-value for the Kasur vs Lahore overall comparison (P = 0.63 I think). (2) Suggest change to “Coxiella burnetii prevalence in the four species of ruminants was”. (3) Suggest change to “although the differences between districts for each species were not statistically different (Table 2). I am not sure if the logistic regression model for looking at species × district interaction adds more, and it will be affected by the sample sizes, but will leave to authors if they decide to retain it. (4) Please add a bit more explanation about abortion as an outcome variable, I wasn’t sure how it was analysed giving an overall P-value of P = 0.03 (not described in Data Analysis section).

Page 6, Table 2: For the more detailed results in Table 2, comparing districts within each species, the Pearson chi-square procedure will fail due to small sample sizes. I would recommend using the Fisher exact test in that instance, the resultant P-values are cattle: 0.30, buffalo: 0.37, goat: 0.39, and sheep: 0.34. They don’t change much apart from goats.

Page 7, Figure 2: By “Labeled sequences”, do you mean the samples with the solid square symbol, i.e. your samples? Please clarify. Also, for publication, you may need higher quality image resolution than in this manuscript.

Page 9, Paragraph 1: Please clarify what the overall aim was: In the Introduction it was stated it was about prevalence estimation, but here it is about molecular characterisation. Also please avoid too much repetition of information already provided in the Introduction.

, Paragraph 2: Suggest change “The previous study” to “A previous study”.

, Paragraph 3: Suggest change “The distribution” to “The estimated prevalence”.

Page 10, Paragraph 1: The description of cattle and buffalo should be in a new paragraph, separate from your discussion of sheep. And then your overall recommendation in a final separate paragraph, which could be expanded with the specifics of recommendations.

References:

[2] Benenson & Tigertt: Missing volume number.

[13] Derrick: Missing page numbers.

[23] Tozer et al: Change to “Queensland”.

[26] Iqbal et al. This has now been published in Pak. J. Zool., please update reference.

6. PLOS authors have the option to publish the peer review history of their article (what does this mean?). If published, this will include your full peer review and any attached files.

Reviewer #1: **Yes: **Ruth Cabrera

Reviewer #2: No

Reviewer #3: No

---

## [Author Response · Author response to Decision Letter 0]

4 Mar 2024

I would like to thank the reviewers very much for the detailed review of the manuscript. I appreciate the insights and believe the manuscript has been improved from their input. The suggestions are incorporated appropriately. The following are the responses to each of the reviewers’ comments:

Reviewer 1

1. For easier review, line numbers should be shown on the left side of each page. Also, please make sure to subject the manuscript to language review and editing.

Ans. Line number has been incorporated. The language of manuscript is reviewed and edited.

2. In the introduction, please don't begin every sentence with C. burnetii, and the way punctuation marks are used confuse the reader. 

Ans. The sentences initiated by the word “C. burnetii” have been rephrased and the sentence is rephrased with edited punctuation marks as per suggestions (Line 46 – 57).

3. Tables in general, including the Supplementary ones are poorly organized and difficult to understand. I can't see the difference between table 1 and 2. And as a recommendation, when there are rows that add up to zero, it means that the sample size was insufficient, and that analyzes carried out are not appropriate. 

4. Ans. In supplementary tables, the table 1 refers to the statistical difference between exotic and indigenous breed while table 2 refers to the statistical difference between exotic and indigenous breed among the studied individual species i.e. cattle, buffalo, goat and sheep. As the Pearson’s chi square test is not accurate for small sample size, Table 2 is re- analyzed using Fisher s’ exact test. All the supplementary tables are re- analyzed using Fisher s’ exact test (Table S2, S3 and S4 were re-analyzed using Fishers exact test, Please review the supplementary tables file).

5. Phylogenetic analysis is insufficient. More isolates, both reported and those derived from their own study should be added.

Ans. More isolates are incorporated in the phylogenetic analysis. One additional query sequence from Lahore and reported sequences from two cities of Turkey and Tamil Nadu city of India 

 (Fig. 2).

6. The discussion is also poorly written. What was already mentioned in the summary, introduction and results is constantly repeated.

Ans. The discussion is rephrased in the edited manuscript. The repeated lines are deleted in the edited manuscript. The information regarding Coxiella burnetii routes of transmission in humans are discussed in the discussion section of the edited manuscript.

Reviewer 2

1. The introduction is informative, but it lacks a clear statement of the research gap and a hypothesis. The authors could improve the introduction by providing more context on the global prevalence of C. burnetii in raw milk and its impact on public health. Furthermore, the reference citations in the introduction are somewhat outdated, and more recent literature should be incorporated to support the study's rationale.

Ans. The research gap and hypothesis statement is incorporated in the edited manuscript (Line 97 -101). The information regarding worldwide prevalence of Coxiella burnetii in raw milk and its implications on public health is concerned in the introduction section of the edited manuscript with incorporated recent references (Line 69 -89).

2. The methodology section lacks details on the inclusion/exclusion criteria for farms and animals, potentially introducing selection bias. The use of a convenient sampling approach may not be representative of the overall population, and the authors should discuss the potential limitations associated with this sampling strategy. Additionally, the absence of information on the sensitivity and specificity of the PCR assay used raises concerns about the reliability of the results.

Ans. The inclusion and exclusion criteria for farms and animals to collect the raw milk samples have been defined in the methodology section with implementing sampling strategy limitations (Line 112 -115). The sensitivity and specificity of the used PCR assay has been discussed in the edited manuscript (Line 138- 141).

3. While the discussion compares the study findings with previous research, it fails to discuss the implications of the results for public health and the dairy industry in Punjab. Additionally, the discussion could benefit from a more critical analysis of the study limitations, such as the potential for contamination during sample collection and the use of a single gene target for detection.

Ans. The implications of Coxiella burnetii infection on public health and dairy industry are concerned in the discussion section of the edited manuscript (Line 278 -281). The limitations regarding gene targets have been discussed in the discussion section (Line 286 – 295).

4. The conclusion provides a succinct summary, but it lacks specific recommendations for future research or practical implications for stakeholders. A more detailed discussion of the policy and public health implications of the study results would enhance the conclusion.

Ans. Recommendations for future concerns for stakeholders and policies for public health concerns have been suggested in the conclusion section of the edited manuscript 

(Line 303 – 309).

5. Additionally, the authors should consider updating references to include more recent publications in the field. Lastly, the manuscript would benefit from a more explicit statement on ethical considerations, especially regarding animal welfare and informed consent for sample collection.

Ans. The manuscript has been proofread for language review. The updated reports have been cited in the introduction and discussion section of the edited manuscript. Ethical statement regarding animal rights and consent has been incorporated in the methodology section of the manuscript (Line 162 – 167).

Reviewer 3

1. There were two aspects investigated: (1) molecular characterising and (2) prevalence estimation, investigating differences between species and geographic districts. The second part was a bit more restricted, in terms of sufficient number of animals across all species and district combinations, but nevertheless this is useful as a reference base for other future studies, and the authors provided a comprehensive analysis of the available data. 

Ans. It is the first evidence study for Coxiella burnetii detection in raw milk samples in Pakistan. In the past studies the Coxiella burnetii was detected in blood, meat and serum samples of livestock animals. There are no studies available on milk samples so far. The present study estimated the Coxiella burnetii prevalence in raw milk samples collected from two districts of Punjab, Pakistan. Future studies will analyze its diagnosis by targeting animals on large scale and other geographic districts of Punjab, Pakistan.

2. One point that needs to be clarified is the overall objective due to a clash between the Introduction and Discussion. I think it really is both points above. In addition, some of the analyses need to be described in a bit more detail in the “Data Analysis” section to clarify some of the results being presented.

Ans. The objective is clarified in the discussion section. The explanatory variables are mentioned in the data analysis section (Line 157-160).

The following is a list of points I noted as I read through the manuscript. Most are minor editorial, but some may take some more time to work through. Please consider each point carefully, particularly the more substantial points, as this will result in a considerably improved manuscript.

3. Page 1: In the title page, I believe the affiliation for Wasim Shehzad should be ‘1’, not ‘2’.

Ans. The affiliation has been replaced to 1 in the title page of the edited manuscript

 (Line 4).

4. Page 2, Abstract: (1) I think “convenience sampling” rather than “convenient”. (2) Also, be consistent in number of decimal places; suggest only one, so “7.4%” and “5.2%”. (3) Change “univariate” to “univariable” (The former indicates only one outcome variable. The latter indicates only one explanatory variable in the model). (4) Change “was differed”to“differed”.

Ans. The word “convenience sampling” is replaced with convenient sampling in the edited manuscript (Line 23). The one decimal point is followed in the manuscript (Line 27). The word “univariate” is replaced by “univariable” (Line 29).

5. Paragraph 3: Suggest “contaminated milk” rather than “contagious milk”.

Ans. The contagious milk is replaced by contaminated milk as per suggestion (Line 53). 

6. Page 3, Paragraph 2: Change “Multicopy” to “multicopy”.

Ans. The word “Multicopy” is replaced by “multicopy” (Line 66).

7. Paragraph 3: Change “about six” to “six” as that is what you have reported. However, please be more specific about “international databank”, e.g. a PubMed search? Make sure this list is current.

Ans. The word “about six” is replaced by “six”(Line 96). The databank is specified in the edited manuscript (Line 97).

8. Paragraph 4: Change “convenient” to “convenience sampling”. Please provide a breakdown of the 304 samples by species.

Ans. The word “convenient” is replaced by “convenience sampling” (Line 106). Individual species sub division is incorporated in the manuscript (Line 107 & 108).

9. Paragraph 5: Change “-20 °C temperature” to “-20 °C”.

Ans. The word “-20 °C temperature” is replaced by “-20 °C” (Line 126). 

10. Page 4, Paragraph 3: For the logistic regression, you need to mention the terms in the model, i.e. explanatory variable(s), not just the outcome variable. Also, where you used a chi-square analysis, is that what you labelled “univariate” (or “univariable” as mentioned above?Pleaseclarify.

Ans. The explanatory variables are incorporated in the edited manuscript (Line 159 & 160). The fisher test function is used in the revised version (Line 157). 

11. Paragraph 4: Perhaps write as “The University of Veterinary and Animal Sciences Advanced Studies and Research Board approved the study at its 50th meeting held on 8 February 2019” (suggesting explicit month name because of differing UK / US date format conventions – or was it “2 August 2019”?

Page 5, Figure 1: Change to “Lahore” in caption.

Ans. The date is formatted as “8 February 2019” in the edited manuscript as per suggestions (Line 164). The word “Lahure” is replaced by “Lahore” in the Figure 1 label (Line 172).

12. Paragraph 1: (1) As in Abstract, please round to 1 d.p., as sample sizes don’t lend to 2 d.p. accuracy. After “not statistically significant’, add P-value for the Kasur vs Lahore overall comparison (P = 0.63 I think). (2) Suggest change to “Coxiella burnetii prevalence in the four species of ruminants was”. (3) Suggest change to “although the differences between districts for each species were not statistically different (Table 2). I am not sure if the logistic model for looking at species × district interaction adds more, and it will be affected by the sample sizes, but will leave to authors if they decide to retain it. (4) Please add a bit more explanation about abortion as an outcome variable, I wasn’t sure how it was analysed giving an overall P-value of P = 0.03 (not described in Data Analysis section).

Ans. I) 1 decimal point is followed in the edited manuscript (Line 177). II) P value is incorporated in the edited manuscript (Line 178). III) The phrase “although the difference was not statistically different” is rephrased as Coxiella burnetii prevalence in the four species of ruminants was” in the edited manuscript (Line 178).

13. Page 6, Table 2: For the more detailed results in Table 2, comparing districts within each species, the Pearson chi-square procedure will fail due to small sample sizes. I would recommend using the Fisher exact test in that instance, the resultant P-values are cattle: 0.30, buffalo: 0.37, goat: 0.39, and sheep: 0.34. They don’t change much apart from goats.

Ans. Fisher exact test is applied in Table 2. The values were same as mentioned above i.e. cattle: 0.30, buffalo: 0.37, goat: 0.39, and sheep: 0.34. The values are incorporated in the edited manuscript (Line 202 - 203).

14. Page 7, Figure 2: By “Labeled sequences”, do you mean the samples with the solid square symbol, i.e. your samples? Please clarify. Also, for publication, you may need higher quality image resolution than in this manuscript.

Page 9, Paragraph 1: Please clarify what the overall aim was: In the Introduction it was stated it was about prevalence estimation, but here it is about molecular characterisation. Also please avoid too much repetition of information already provided in the Introduction.

Ans. The samples with squared symbols were query sequences. Discussion and introduction sections are updated. The discussion section is rephrased to avoid repetition in the edited manuscript.

15. Paragraph 2: Suggest change “The previous study” to “A previous study”. Paragraph 3: Suggest change “The distribution” to “The estimated prevalence”.

Ans. The word “The previous study” is replaced to “A previous study” (Line 251). The word “The distribution” is replaced to “The estimated prevalence” in the edited manuscript (Line 258).

16. Page 10, Paragraph 1: The description of cattle and buffalo should be in a new paragraph, separate from your discussion of sheep. And then your overall recommendation in a final separate paragraph, which could be expanded with the specifics of recommendations.

References:

[2] Benenson & Tigertt: Missing volume number.

[13] Derrick: Missing page numbers.

[31] Tozer et al: Change to “Queensland”.

[35] Iqbal et al. This has now been published in Pak. J. Zool., please update reference.

Ans. The references are updated in the edited manuscript as per suggestion.

---

## [Decision Letter · Decision Letter 1]

19 Mar 2024

Molecular identification of Coxiella burnetii in raw milk samples collected from farm animals in districts Kasur and Lahore of Punjab, Pakistan

PONE-D-23-27370R1

Dear Dr. Muhammad Yasir Zahoor,

We’re pleased to inform you that your manuscript has been judged scientifically suitable for publication and will be formally accepted for publication once it meets all outstanding technical requirements.

Kind regards,

Gianmarco Ferrara, PhD, MVD

Academic Editor

PLOS ONE

Additional Editor Comments (optional):

Reviewers' comments:

Reviewer's Responses to Questions

**Comments to the Author**

1. If the authors have adequately addressed your comments raised in a previous round of review and you feel that this manuscript is now acceptable for publication, you may indicate that here to bypass the “Comments to the Author” section, enter your conflict of interest statement in the “Confidential to Editor” section, and submit your "Accept" recommendation.

Reviewer #1: All comments have been addressed

Reviewer #3: All comments have been addressed

2. Is the manuscript technically sound, and do the data support the conclusions?

Reviewer #1: Yes

Reviewer #3: (No Response)

3. Has the statistical analysis been performed appropriately and rigorously? 

Reviewer #1: Yes

Reviewer #3: (No Response)

4. Have the authors made all data underlying the findings in their manuscript fully available?

Reviewer #1: Yes

Reviewer #3: (No Response)

5. Is the manuscript presented in an intelligible fashion and written in standard English?

Reviewer #1: Yes

Reviewer #3: (No Response)

6. Review Comments to the Author

Reviewer #1: After review, I consider the article suitable for publication. However, as a last comment, the figures still do not have the necessary resolution, and are difficult to view.

Reviewer #3: (No Response)

7. PLOS authors have the option to publish the peer review history of their article (what does this mean?). If published, this will include your full peer review and any attached files.

Reviewer #1: No

Reviewer #3: No

---

## [Editor Report · Acceptance letter]

26 Apr 2024

PONE-D-23-27370R1 

PLOS ONE

Dear Dr. Zahoor, 

I'm pleased to inform you that your manuscript has been deemed suitable for publication in PLOS ONE. Congratulations! Your manuscript is now being handed over to our production team.

Kind regards, 

on behalf of

Dr. Gianmarco Ferrara 

Academic Editor

PLOS ONE